# Improvement of Ulcerative Colitis by Aspartate via RIPK Pathway Modulation and Gut Microbiota Composition in Mice

**DOI:** 10.3390/nu14183707

**Published:** 2022-09-08

**Authors:** Xian Hu, Xinmiao He, Can Peng, Yiwen He, Chenyu Wang, Wenjie Tang, Heshu Chen, Yanzhong Feng, Di Liu, Tiejun Li, Liuqin He

**Affiliations:** 1Hunan Provincial Key Laboratory of Animal Intestinal Function and Regulation, Laboratory of Animal Nutrition and Human Health, College of Life Sciences, Hunan Normal University, Changsha 410081, China; 2Key Laboratory of Agro-Ecological Processes in Subtropical Region, Hunan Provincial Key Laboratory of Animal Nutritional Physiology and Metabolic Process, Institute of Subtropical Agriculture, Chinese Academy of Sciences, Changsha 410125, China; 3Heilongjiang Academy of Academy of Agricultural Sciences, Harbin 150086, China; 4Animal Breeding and Genetics Key Laboratory of Sichuan Province, Sichuan Animal Science Academy, Chengdu 610066, China

**Keywords:** aspartate, intestinal immunity, ulcerative colitis, gut microbiota, RIPK pathway

## Abstract

The intestine requires a great deal of energy to maintain its health and function; thus, energy deficits in the intestinal mucosa may lead to intestinal damage. Aspartate (Asp) is an essential energy source in the intestinal mucosa and plays a vital part in gut health. In the current study, we hypothesized that dietary supplementation of Asp could alleviate DSS-induced colitis via improvement in the colonic morphology, oxidative stress, cell apoptosis, and microbiota composition in a mouse model of dextran. Asp administration decreased the disease activity index, apoptosis, myeloperoxidase, eosinophil peroxidase, and proinflammatory cytokine (IL-1β and TNF-α) concentrations in the colonic tissue, but improved the body weight, average daily food intake, colonic morphology, and antioxidant-related gene (GPX1 and GPX4) expression in DSS-treated mice. Expression levels of RIPK1 and RIPK3 were increased in the colon following Asp administration in the DSS-induced mice, whereas the MLKL protein expression was decreased. 16S rRNA sequencing showed that Asp treatment increased the abundance of *Lactobacillus* and *Alistipes* at the gene level, and *Bacteroidetes* at the phylum level, but decreased the abundance of *Actinobacteria* and *Verrucomicrobia* at the phylum level. Asp may positively regulate the recovery of DSS-induced damage by improving the immunity and antioxidative capacity, regulating RIPK signaling and modulating the gut microbiota composition.

## 1. Introduction

Ulcerative colitis (UC), a recurrence and relieving inflammatory bowel disease (IBD), influences the integrity of the mucosa and submucosa and the function of the rectum and colon [1]. Although the pathogenesis of IBD has not been definitively elucidated, factors including genetic mutations, immunological disorders, environmental exposure, oxidative stress, and intestinal flora have been suggested to be involved in this process [2]. Notably, cell necroptosis is also a considerable feature of epithelial barrier disorders in UC, which aggravates secondary injuries [3]. To date, conventional treatments for UC have mainly focused on relieving clinical symptoms. First-line therapy drugs such as amino salicylate, immunosuppressive agents, and corticosteroids cannot achieve a satisfactory therapeutic effect, while metabolic and immune disorders occur frequently [4]. Recently, nutritional therapies such as probiotics, amino acids, and plant extract supplement administration have received more attention due to there being fewer side effects [5].

It has been reported that amino acids play a pivotal role in the clinical care of all patients with UC [6]. For example, aspartate (Asp), one of the main energy sources in the intestinal mucosa, maintains intestinal barrier integrity and regulates nucleotide, protein, lipid, and energy metabolism [7,8]. Some studies have found that Asp pretreatment can alleviate oxidative stress in myocardial infarction (MI)-induced rats by increasing the levels of glutathione and mitochondrial adenine nucleoside triphosphate (ATP) while decreasing the level of lipid peroxides [9]. Other studies have suggested that Asp supplementation attenuates endoplasmic reticulum stress and cell apoptosis in trinitro-benzene-sulfonic acid-induced colitis [10]. 

In addition, recent studies have reported that the receptor-interacting protein kinase (RIPK) pathway stimulated by the tumor necrosis factor family of cytokines plays a large part in modulating cell death or apoptosis and plays a key role in intestinal inflammation and cell survival, especially in UC pathophysiologies [11]. Weinlich et al. (2015) reported that receptor interacting protein kinase 1 (RIPK1) could maintain intestinal function by suppressing receptor interacting protein kinase 3 (RIPK3)-mixed-lineage kinase domain-like protein (MLKL)-dependent necroptosis and caspase 8-mediated apoptosis in multiple tissues and cell types such as keratinocytes, oncocytes, and colonic epithelial cells [12]. However, there has been no study on the effect of exogenous Asp on the regulation of the RIPK pathway associated with the immune and antioxidant systems in the UC model. Thus, the present study was conducted to investigate whether rectal perfused Asp could improve the immunity and antioxidant capacity, cell apoptosis, and intestinal flora by regulating the RIPK pathway in dextran sulfate sodium (DSS)-induced colitis in mice.

## 2. Materials and Methods

### 2.1. Animals and Experimental Design

Male C57BL/6J mice (10 weeks old) were purchased from Slac Laboratory Animal Central (Changsha, China) All mice were housed under constant conditions (room temperature 24 ± 1 °C, 12 h light/dark cycle, lights off at 20:00) with provided free access to food and water. This study was approved by the animal welfare committee of the Institute of Subtropical Agriculture, Chinese Academy of Sciences (2013020; Changsha, China).

After a 1-week adaptation period, 40 mice were randomly assigned to five groups (*n* = 8/group) (Figure 1A). During the first 7 days, mice in the control (CON) group received only water, while mice in the DSS-treated (DSS, MW 36,000–50,000 Da, MP Biomedicals, Irvine, CA, USA), low-dose Asp-treated (LA, 0.15 mM) (Asp purity >99 %; Amino Acid Bio-Chemical, Wuhan, China), medium-dose Asp-treated (MA, 2.3 mM), and high-dose Asp-treated (HA, 7.7 mM) groups were administered 3.5% DSS in drinking water. On day 8, mice in the CON and DSS groups were only rectally administered PBS, while mice in the LA, MA, and HA groups were administered different doses of Asp (dissolved in PBS) in rectal perfusion for seven consecutive days. Body weight, DAI score (Table 1), and feed intake were monitored daily throughout the experiment. Mice were sacrificed by cervical dislocation, during which blood was collected, and colon samples were collected for further analysis at the end of the experiment.

### 2.2. Histological Analysis

Colon tissue samples were processed and embedded in paraffin using standard techniques. All tissue samples were cut into 10 µm sections for staining with hematoxylin and eosin (H&E) and performed by two blinded investigators to the experimental groups. The extent of inflammation was measured and scored as described previously [13] (Table 2).

### 2.3. Quantification of Myeloperoxidase and Eosinophil Peroxidase Activity and Inflammatory Cytokines Contents in the Colon

Colon tissue samples were homogenized in a buffer containing protease inhibitors for 1 min and centrifuged at 12,000 rpm for 10 min at 4 °C. The supernatant were obtained and stored at −80 °C for subsequent measurement. The myeloperoxidase (MPO) and eosinophil peroxidase (EPO) activity and the concentrations of inflammatory cytokines tumor necrosis factor-α (TNF-α), interleukin-6 (IL-6), and interleukin-1β (IL-1β) in the colon were determined using ELISA quantitative kits (Cusabio Biotech, Wuhan, China) according to the manufacturer’s instructions.

### 2.4. RT-qPCR Analysis

Total RNA was extracted from the colonic samples using TRIzol (AG, Changsha, China. The Prime Script RT Reagent Kit (Takara, Dalian, China) was used to obtain cDNA, quantitative real-time polymerase chain reaction (RT-qPCR) was performed with SYBR R Premix Ex TaqTMII RT-qPCR was performed in 10 μL of assay volume containing 5 μL of SYBR Green mix (Takara), 3 μL of diethyl pyrocarbonate-treated deionized H_2_O, 0.2 μL of ROX, 1 μL of the cDNA template, 0.4 μL of the forward primer, and 0.4 μL of the reverse primer. The relative amount of the target mRNA was normalized to the β-actin level, and the results were calculated by the 2^−^^ΔΔCt^ method. The primer sequences are shown in Table 3.

### 2.5. Protein Qualification Using the Wes Simple Western System

Total protein was extracted from the colon samples using RIPA lysis buffer with buffer supplemented with a protease and 1% PMSF. The BCA Assay Kit was used to determine the protein concentration. Equal numbers of protein (50 µg) were separated by 10% SDS-PAGE at 100 V for 2 h and transferred to a nitrocellulose membrane using a semi-dry apparatus. The membranes were blocked with 5% non-fat milk for 2 h at ±20 °C and incubated overnight at 4 °C with the following primary antibodies: RIPK1, RIPK3, and MLKL at a 1:500. The next day, the membranes were washed with PBST buffer and incubated with appropriately diluted horseradish peroxidase-conjugated secondary antibodies for 1 h at room temperature after washing with TBST three times. The protein bands were visualized using an ECL kit on a (Bio-Rad ChemiDoc XRS+ image analyzer (Bio-Rad, Hercules, CA, USA) after TBST washing, as previously described. β-actin was used as an internal reference.

### 2.6. Terminal Deoxynucleotidyl Transferase Mediated dUTP Nick End Labelling (TUNEL) Assay

The Promega Dead End Colorimetric TUNEL System Kit (Roche, Shanghai, China) was used to stain apoptotic nuclei in colon tissue to detect cell apoptosis according to the manufacturer’s instructions. When cell apoptosis occurs, endogenous nucleases are activated, causing double-strand breaks or gaping in one strand of DNA, resulting in a series of 3′-OH ends. DNA cuts in tissue cells can be labeled with biotin-dUTP, namely, TUNEL can specifically bind to horseradish peroxidase streptavidin (streptavidin-HRP) (Roche, Shanghai, China). In the presence of (DAB), the display is brown (peroxidase activity), which specifically and accurately locates the apoptotic cells, and can observe and count the apoptotic cells under a normal microscope; because there were almost no live or proliferating cell DNA breaks, there was no 3′-OH formation, and rarely stained.

### 2.7. Transmission Electron Microscopy (TEM)

The colonic tissues of the experimental mice were collected and washed with cold phosphate buffered saline. The colonic segments were fixed using 3% malondialdehyde, and then dehydrated by acetone. The dehydrated tissue was infiltrated by a dehydrating agent and epoxy resin (Epon812) in turn and then filled with embedding solution. The samples were embedded in epon-araldite resin and ultrathin sectioned at 50 nm, and the sections were stained with uranyl acetate and lead citrate.

### 2.8. Taxonomic Analyses of the Gut Microbiota

Terminal colonic chyme were stored at −80 °C after being snap frozen in liquid nitrogen until analysis. Microbial DNA was extracted from the fecal samples. The V3–V4 hypervariable regions of the bacterial 16S rRNA gene were amplified with the primers 338F using a thermocycler PCR system (Gene Amp 9700, ABI, Waltham, MA, USA) The sequences were clustered into function taxonomic units of at the very least 97% identity, and the correlative abundances of the microbial taxa (genus to kingdom) were generated from nonrarefied operational taxonomic unit tables. Species richness (alpha diversity) was measured using the Chao1 index. Beta diversity was calculated using the UniFrac distance between samples and visualized in three-dimensional plots according to a weighted principal coordinate analysis. A linear discriminant analysis effect size (LEfSe) analysis was accomplished to identify significantly different phylotypes among the experimental groups. In addition, the prediction of flora function was based on FAPROTAX platform (Novogene, Beijing, China).

### 2.9. Statistical Analysis

Statistical analysis was executed using one-way ANOVA using the data statistics software (SPSS, SPSS 22.0, Chicago, IL, USA). All measurement data are expressed as the mean ± standard error (SEM). The SPSS statistical analysis histogram was used to analyze the data distribution, and the test data conformed to the normal distribution. At the same time, we used a post hoc test after ANOVA. Statistical significance was set at *p* < 0.05.

## 3. Results

### 3.1. Effects of Asp on Growth Status, Colonic Morphology, and Pathology in DSS-Induced Mice

To establish a model of colitis, we treated mice with 3.5% DSS under free drinking water for 7 days. We found that compared with the CON group, on day 3, the average body weight (BW) and average daily feeding intake (ADFI) was highly significantly decreased in the DSS group, and on day 7, these values reached their lowest values (*p* < 0.001). Furthermore, the disease activity index (DAI) score of the DSS-induced mice was the highest and was accompanied by hematochezia in the stool. On day 8, when mice were rectally infused with different levels of Asp, the results showed that, compared with the DSS group, the BW in the LA, MA, and HA groups began to increase as the Asp treatment time increased. On day 10, there was no significant difference in the BW between the LA and CON groups (*p* > 0.05). On day 12, the BW in the LA and HA groups was different from that in the DSS group. As for the whole experiment, compared with the CON group, ADFI in the DSS and MA groups significantly decreased (*p* < 0.01); however, the values in the LA and HA groups showed no difference (Figure 1).

Colon length shortening was positively correlated with inflammation and edema in the DSS-induced colitis mice. The colon length in the DSS group was significantly shorter than that in the CON group, and the dark red colons were accompanied by swollen, bleeding intestinal walls, resulting in a higher DAI score, while the Asp treatments at 0.15, 2.3, and 7.7 mM evidently prolonged the colon length and reduced the DAI score. As for the colonic morphological structure, the CON group had intact mucous membranes and neat villi with healthy crypt structures (Figure 1C,E), which were enriched in goblet cells without inflammatory cell infiltration or mucosal erosion. The DSS-induced colitis mice showed mucosal and submucosal edema, severe inflammatory cell infiltration, crypt loss, and epithelial injury. However, treatment with 0.15, 2.3, and 7.7mM doses of Asp significantly decreased the histological scores with the manifestation of well-preserved crypt structures and intestinal mucosal integrity compared with the DSS group (Figure 1F).

### 3.2. Effects of Asp on Concentration of Inflammatory Cytokines and Expression of Inflammation Related-Genes in the Colon of DSS-Induced Mice

To evaluate the effect of Asp on inflammatory responses in the colon of DSS-induced mice, the concentrations of colonic pro-inflammatory cytokines (IL-1β, TNF-α, and IL-6) were measured. The results showed that compared with the CON group, the levels of TNF-α, IL-1β, and IL-6 in the DSS group significantly increased (Figure 2D–F). Compared with the DSS group, the levels of TNF-α and IL-6 in the HA group showed a decreasing trend, and the level of IL-1β in the HA group significantly decreased, but there was no significant difference compared with the CON group. As for the mRNA expression of inflammation related genes compared to the CON group, the mRNA expression level of TNF-α in the DSS group significantly increased by 52%, while TLR4 and MYD88 mRNA abundance increased by 13.9% and 24.7%, respectively (Figure 2A,B). However, the mRNA expression of TNF-α in the HA and MA groups significantly decreased compared to that in the DSS group. The mRNA expression levels of TLR4 and MYD88 in the HA group were lower than those in the DSS group (Figure 2A,B). These results suggest that a high dose of Asp has an anti-inflammatory effect in the DSS-induced colitis mouse model.

### 3.3. Effects of Asp on Intestinal Mitochondrial Ultrastructure and Antioxidant Indexes in the Colon ff DSS-Induced Mice 

To explore the effect of Asp on the intestinal ultrastructure integrity in colitis, we focused on mitochondrial damage using TEM. We found that DSS treatment resulted in varying degrees of mitochondrial damage. Compared with the CON group, the mitochondria of the colon were heavily swollen, and the number of mitochondria significantly decreased in the DSS group, and the structure and number of mitochondria in the LA, MA, and HA groups were better than those in the DSS group to different degrees. In particular, 2.3 or 7.7 mM-Asp treatment could markedly maintain the structural integrity of the mitochondria and increased the mitochondrial number compared to the DSS group, and the 0.15-mM Asp treatment could alleviate the mitochondria damage to some degree (Figure 3A). 

It is well-known that MPO and EPO activities can determine the degree of neutrophil infiltration and oxidative stress situation [14]. The results showed that the MPO and EPO activities were higher in the DSS group than those in the CON group. In contrast, the MPO and EPO activities in the HA group decreased by 30% compared with those in the DSS group (Figure 3B,C). Moreover, for the antioxidative associated-genes, the DSS treatment resulted in a decrease in the mRNA expression of GPX1 and GPX4 in the colon (Figure 3D,E). There was no difference in the mRNA expression of GPX1 among these groups; however, the mRNA expression level of colonic GPX4 in the HA group was the highest and showed a significant difference compared with the other groups (*p* < 0.05).

### 3.4. Effects of Asp on Cell Apoptosis and Expression of RIPK Pathway in the Colon of DSS-Induced Mice

To further confirm whether Asp administration could alleviate the apoptosis and necrosis rate in colonic epithelial cells of DSS-induced mice, we detected cell apoptosis using the TUNEL method and expression of the necrosis-associated key pathway using real-time PCR and Western blotting. The results showed that the cell apoptosis rate in the DSS group was higher than that in the other groups, but the cell apoptosis rate significantly decreased with increasing concentrations of Asp (Figure 4A). The DSS treatment increased the mRNA expression levels of RIPK1, MLKL, and RIPK3 with or without Asp administration, and the mRNA expression of MLKL and RIPK3 in the DSS group was higher than that in the CON group (Figure 4C,D). However, compared to the DSS group, rectal infusion of high doses of Asp increased the mRNA expression of MLKL and RIPK3, but decreased the mRNA expression level of RIPK1 in the colon (Figure 4B). In addition, we evaluated the activation of the RIPK1, RIPK3, and MLKL proteins by Western blot analysis. Compared with the CON group, the protein expression of RIPK1, RIPK3, and MLKL in the DSS group increased, and the protein expression of RIPK1 and RIPK3 in the colon increased under Asp administration at different levels, whereas the MLKL protein expression was reduced, especially in the MA and HA groups (Figure 4E–G).

### 3.5. Effects of Asp on Colonic Microbes in DSS-Induced Mice 

The colonic microbes were analyzed through the sequencing of 16S rRNA genes, and the results showed that there were no significant differences in Chao, Shannon, and Simpson among these five groups, but the DSS treatment led to a decrease in the community evenness, richness, and diversity to a certain extent (Figure 5). 

At the phylum level, *Bacteroidetes* and *Firmicutes* mainly accounted for 75% and 18%, respectively, and 10 dominant microbes were changed in response to Asp treatment (Figure 6A). Compared to the CON group, the abundance of *Bacteroidetes* decreased, but *Actinobacteria* and *Verrucomicrobia* increased in the DSS group. However, compared with the DSS group, Asp administration decreased *Actinobacteria* and *Verrucomicrobia* but increased *Bacteroidetes* (Figure 7A–C). At the genus level, compared with the CON group, the abundance of *Lactobacillus* and *Alistipes* decreased in the DSS group (Figure 6B), but *Akkermansia* abundance significantly increased (Figure 7D–F). *Lactobacillus* abundance in the HA group was the highest, but the abundance of *Akkermansia* was lower than that in the DSS group. Functional profiling of the microbial communities was predicted using the FAPROTAX method, with key KEGG pathways such as ureolysis, chemoheterotrophs, nitrate reduction, fermentation, nitrate-respiration, and aerobic-chemoheterotrophs (Figure 5C).

## 4. Discussion

Chronic DSS exposure is a common model used to investigate the mechanisms of UC. In this model, the excessive formation of inflammatory cytokines and reactive oxygen species is considered as a trigger for the initiation of colitis [15]. In the present study, our results showed that under the DSS treatment conditions, the clinical symptoms of mice including BW loss, decreased ADFI, diarrhea, fecal blood, and colon shortening suggested that the colitis model was successfully established. When we treated these colitis mice with rectal infusion of different levels of Asp, the BW and ADFI significantly increased, and the clinical symptoms were also remarkably reduced, in particular, 0.15 and 7.7 mM Asp treatment exhibited a superior effect to improve growth status. It is possible that Asp is conducive to maintain colonic morphology and promote colonic function repair in DSS-induced mice by increasing the ADFI. The results of colon biopsy morphology and mitochondrial ultrastructure confirmed this hypothesis. Our results were consistent with a previous report that dietary supplementation with 0.5 or 1% Asp could improve the growth performance and intestinal morphology in lipopolysaccharide (LPS)-challenged weaned piglets [16].

Abnormal secretion or overexpression of inflammatory cytokines is considered as an important pathogenesis leading to the occurrence and development of colitis [17]. When the colonic epithelial cells are damaged, macrophages are activated and release a large number of inflammatory cytokines including IL-1β, IL-6, and TNF-α, thereby causing intestinal inflammatory responses. In the present study, the results showed that DSS treatment increased inflammatory cytokine secretion, while rectal infusion of 7.7 mM Asp significantly decreased the colonic IL-1β secretion and TNF-α mRNA expression in the DSS-induced mice. In addition, to some degree, different levels of Asp administration decreased the gene expression of TLR4 and MYD88 as well as the TNF-α and IL-6 contents in the UC mice, suggesting that Asp could alleviate colitis in mice by regulating the secretion and expression of inflammatory cytokines. This is further confirmed by a previous study that Asp supplementation increased the intracellular glutamine level and promoted the synthesis of purine and pyrimidine nucleotides in lymphocytes and leukocytes [18,19]. Dietary Asp supplementation has been reported to decrease mRNA expression of the intestinal TLR4-related genes linearly or quadratically in LPS-challenged pigs [8]. Other studies also reported that LPS treatment significantly stimulated the production of TNF-α and IL-6 in RAW 264.7 cells, while the addition of Asp inhibited their secretion [20]. These results are consistent with our results and further confirm that Asp plays a vital role in improving intestinal immunity.

UC is characterized by inflammation and oxidative stress in the rectal and colonic mucosa [14]. Recent studies have shown that dietary Asp supplementation could attenuate the oxidative stress response and intestinal barrier dysfunction in H_2_O_2_-challenged piglets by increasing the expression and activity of catalase and the GPX1/4 gene [21], in which Asp reversed the expression of catalase and the GPX1 gene in the LPS-induced oxidative injury of ovine intestinal epithelial cells and upregulated catalase and GPX gene abundance in young grass carp [22]. This is consistent with our finding that 7.7 mM Asp administration maintained the morphological structure and number of colonic mitochondria and upregulated the gene expression of GPX1 and GPX4 in the DSS-induced UC mice, thereby improving the antioxidant capacity of UC mice. Furthermore, in this study, the activity of MPO and EPO was reduced by Asp treatment to some degree. This further suggests that Asp could effectively inhibit neutrophil infiltration and oxidative stress to mitigate the symptoms of colitis.

Apoptosis and necrosis of the intestinal epithelial cells are significant features of various intestinal diseases [17], especially in UC [23]. RIPK1, as a major regulatory factor, plays a key role in cell survival, inflammation, and apoptosis by regulating the caspase 8-mediated apoptosis pathway [24] and the RIPK3-MLKL-dependent necroptosis pathway [25]. Previous studies have reported that RIPK1 inhibits cell apoptosis and necroptosis through kinase-independent functions, which is important for late embryonic development and the prevention of intestinal inflammation [11]. Further studies have confirmed that RIPK3 facilitated LPS-induced inflammation by activating dendritic cells or macrophages to release mature IL-1β and RIPK3-dependent cell death, resulting in a dampening of TNF-induced pro-inflammatory cytokine/chemokine secretion, thereby leading to a less inflammatory outcome [24]. In the current study, the results showed that Asp treatment decreased the apoptosis rate of colonic epithelial cells and the protein expression of MLKL in the colon of UC mice, while it increased the protein expression of RIPK1 and RIPK3, suggesting that the rectal infusion of different levels of Asp could effectively inhibit cell apoptosis and necrosis caused by colitis. Notably, the colonic RIPK3/MLKL signaling was remarkably downregulated by 7.7 mM Asp administration, indicating the inhibition of MLKL-mediated cell necrosis. However, Asp could activate RIPK1/RIPK3 signaling to alleviate colonic inflammation and oxidative stress, thereby promoting intestinal injury repair.

Gut microbes are known to be closely related to the pathogenesis of UC [14], and intestinal inflammation can drive the loss of microbiota diversity, leading to a distinct microbial community composition [26]. In healthy mice, the microflora is mainly composed of *Firmicutes* and *Bacteroidetes*. *Firmicutes* can provide additional energy for the host by fermenting polysaccharides to short-chain fatty acids (SCFAs), and *Bacteroidetes* can promote an intimate interaction between the microbiota and host by consuming glycans released from the mucus [16]. Previous studies have indicated that dietary Asp supplementation contributes to the modulation of immune responses by decreasing the ratio of *Firmicutes* to *Bacteroidetes* in the feces of weaned piglets [16,27]. However, our results showed that Asp administration increased the ratio of *Bacteroidetes* to *Firmicutes* in the UC mice. One potential explanation for this may be that inflammation caused by colitis is a high energy consuming process during which both energy harvest and storage decreased, thus Asp administration could access the tricarboxylic acid cycle to provide additional energy, thereby changing the proliferation of *Bacteroides* and *Firmicutes*. Furthermore, changes in the relative abundance of *Bacteroidetes* and *Firmicutes* can affect the energy balance. *Firmicutes* is related to energy harvest and storage, while *Bacteroidetes* has the capacity to consume energy [28]. Derrien et al. (2008) suggested that *Akkermansia* could interact with intestinal epithelial cells to produce IL-8 for immunomodulatory effects [29]. Additionally, *Akkermansia* can enhance the intestinal barrier function by enhancing the adhesion and integrity of the intestinal epithelium [30]. It has been reported that an increase in *Lactobacillus* could improve the gut barrier function in response to inflammation. Wu et al. (2019) reported that *Alistipes* abundance was negatively correlated with the DAI, pathological score, and TNF-α, IL-6, and IL-1β levels in UC mice [2]. Our results showed that the abundance of *Lactobacillus* and *Alistipes* decreased under DSS treatment and the abundance of Akkermansia increased in colonic feces, while Asp administration could reverse the values in UC mice. In particular, for the 0.15 or 2.3 mM Asp treatment, DSS may cause epithelial cell damage, thus damaging the gut barrier function, which increased the abundance of *Akkermansia* to fight against the DSS damage. Moreover, gut microbiota can regulate the expression of intestinal TLRs, NOD, and Myd88, which are associated with host inflammation [31]. Our results also confirmed this. The above results indicate that Asp administration might promote colonic repair by upregulating the relative abundance of beneficial bacteria (e.g., Bacteroides, Lactobacillus, and Alistipes).

## 5. Conclusions

In conclusion, our results revealed that Asp administration improved colonic integrity and mitochondrial function, increased the expression and activity of antioxidant enzymes, decreased cell apoptosis, necrosis, and inflammatory cytokine secretion, and altered the composition of the colonic microbiota and their functional profiles to reverse DSS-induced colitis in mice. Moreover, Asp may exert these effects by serving as a regulator of RIPK1 activation and inhibiting the RIPK3/MLKL-mediated necroptosis pathway. These findings provide a new practical solution for treating a range of intestinal diseases and support the use of Asp as a prophylactic agent to prevent the development of IBD. 

## Figures and Tables

**Figure 1 nutrients-14-03707-f001:**
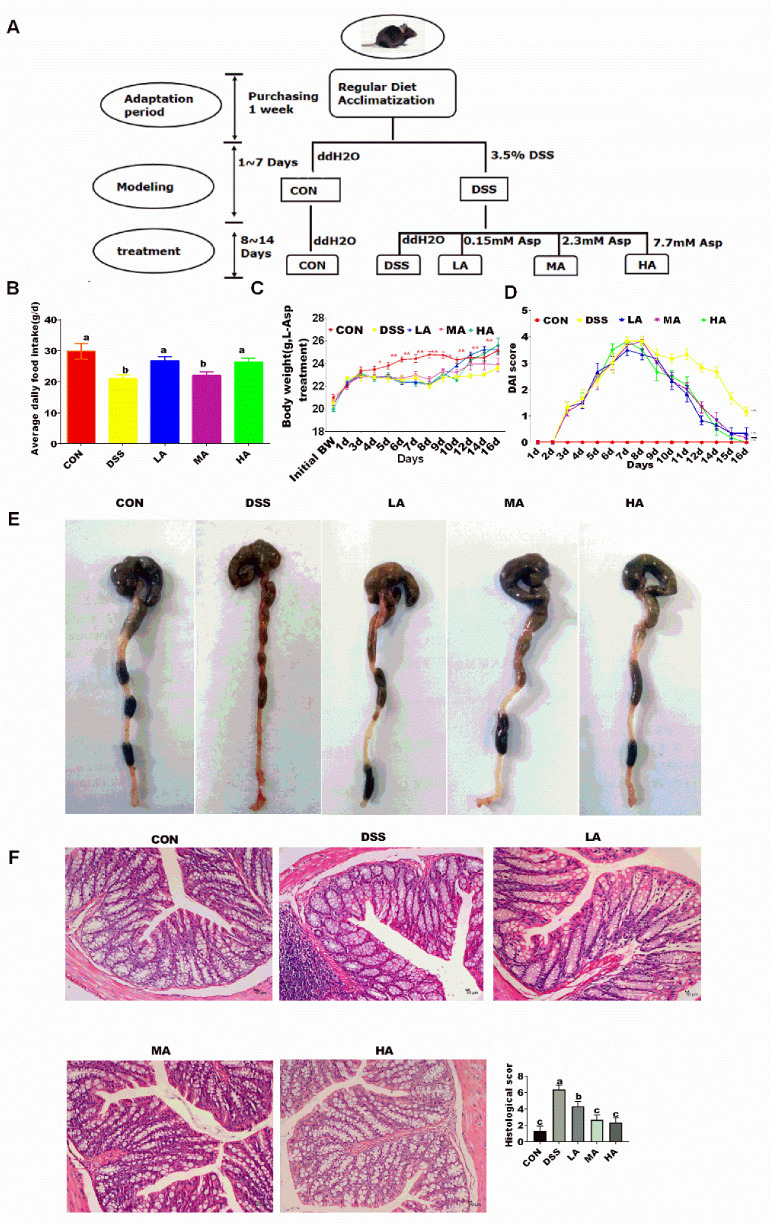
The effects of aspartate on the growth status, colonic morphology, and pathology in the DSS-induced mice. (**A**) Schematic diagram of the experimental design. (**B**) The average daily feed intake. (**C**) Body weight. (**D**) DAI score. (**E**) Macroscopic photographs of the colon lengths. (**F**) Colonic morphology. Values are expressed as the mean ± SEM, *n* = 6. a, b, c Mean values with different small letter superscripts mean a significant difference (*p* < 0.05).

**Figure 2 nutrients-14-03707-f002:**
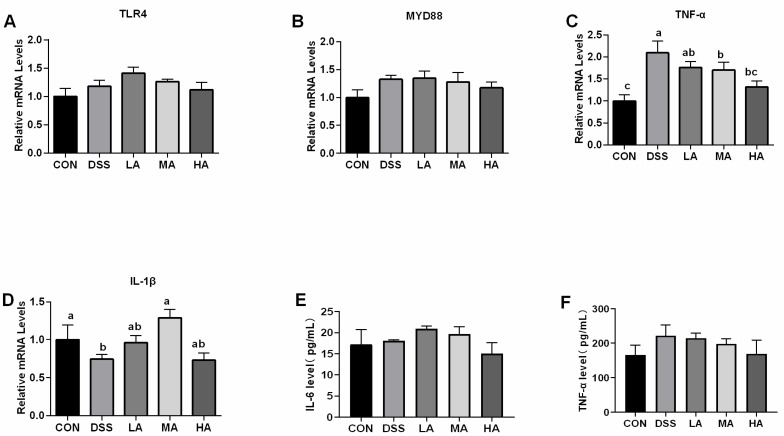
The effects of aspartate on the concentration of inflammatory cytokines and the expression of inflammation related-genes in the colon of the DSS-induced mice. (**A**) The mRNA expression level of colonic TIR4. (**B**) The mRNA expression level of colonic MYD88. (**C**) The mRNA expression level of colonic TNF-α. (**D**) The mRNA expression level of colonic IL-1β. (**E**) The content of IL-6. (**F**) The content of TNF-α. Values are expressed as mean ± SEM, *n* = 6. a, b, c Mean values with different small letter superscripts mean a significant difference (*p* < 0.05).

**Figure 3 nutrients-14-03707-f003:**
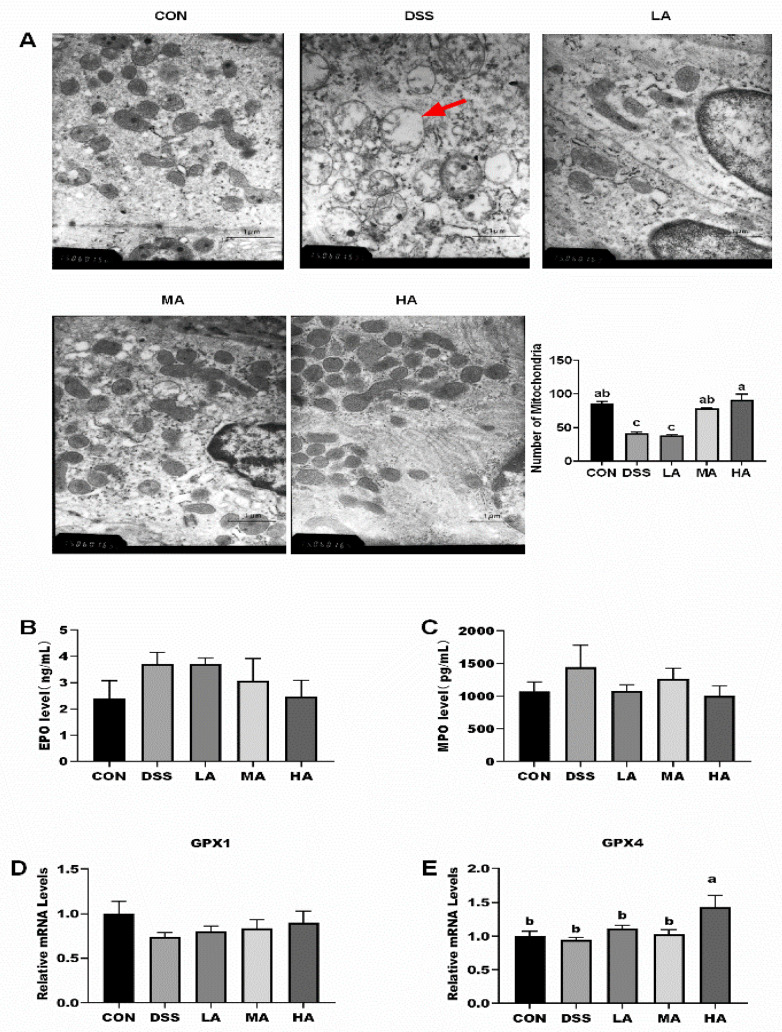
The effects of aspartate on intestinal mitochondria ultrastructure and antioxidant indices in the colon of the DSS-induced mice. (**A**) The morphology and number of colonic mitochondria, black oval mitochondria. (**B**) The content of MPO. (**C**) The content of EPO. (**D**) The mRNA expression level of colonic GPX1. (**E**) The mRNA expression level of colonic GPX4. Values are expressed as the mean ± SEM, *n* = 6. a, b, c Mean values with different small letter superscripts mean a significant difference (*p* < 0.05).

**Figure 4 nutrients-14-03707-f004:**
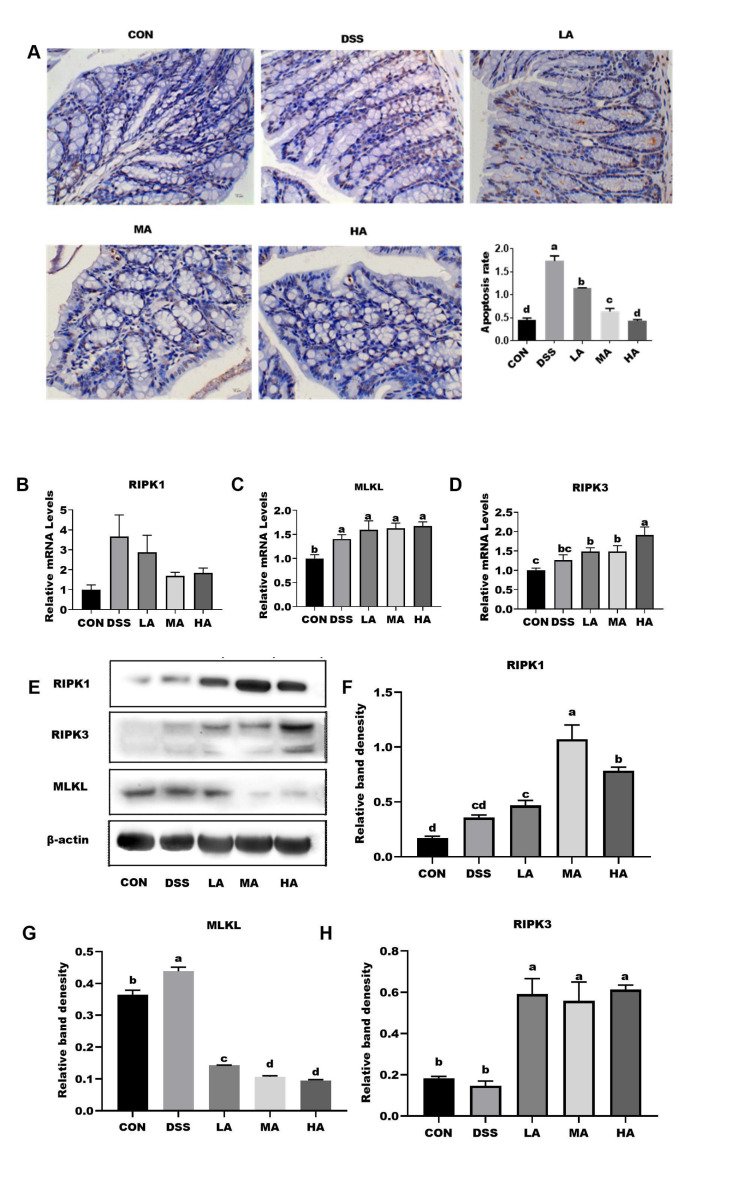
The effects of aspartate on cell apoptosis and the expression of RIPK pathways in the colon of DSS-induced mice. (**A**) The TUNEL assay shows colonic apoptosis section and rate of apoptosis; blue, living cells; brown, dead cells. (**B**) The mRNA expression level of colonic RIPK1. (**C**) The mRNA expression level of colonic MLKL. (**D**) The mRNA expression level of colonic RIPK3. (**E**) The protein band of RIPK1, MLKL, RIPK3, and β-actin. (**F**) The relative protein expression of RIPK1. (**G**) The relative protein expression of MLKL. (**H**) The relative protein expression of RIPK3. Values are expressed as the mean ± SEM, *n* = 6. a, b, c, d Mean values with different small letter superscripts mean a significant difference (*p* < 0.05).

**Figure 5 nutrients-14-03707-f005:**
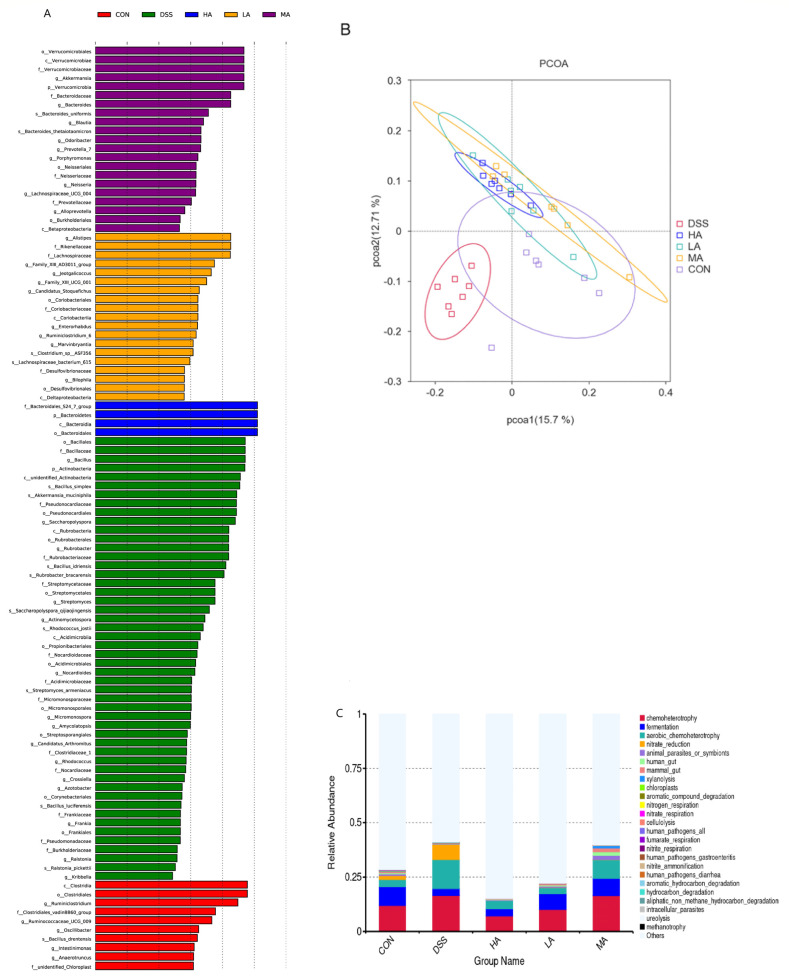
The PCOA figure and predictive functional profiling of colonic microbes. (**A**) Significant flora changes among the CON, DSS, LA, MA, HA groups, as measured by LEfSe analysis. (**B**) The PCOA figure of colonic microbes. (**C**) The predictive functional profiling of microbial communities. Values are expressed as the mean ± SEM, *n* = 6.

**Figure 6 nutrients-14-03707-f006:**
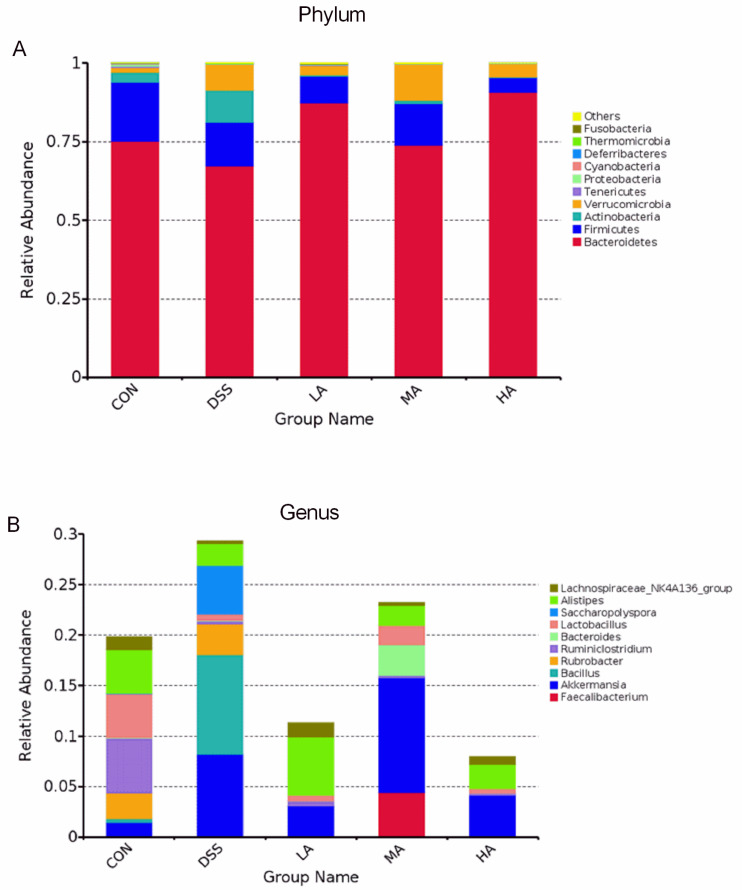
The OTU cluster histogram of the top 10 species with the greatest abundance. (**A**) The top 10 species with the greatest abundance of taxa at the phylum level. (**B**) The top 10 species with the greatest abundance of taxa at the genus level.

**Figure 7 nutrients-14-03707-f007:**
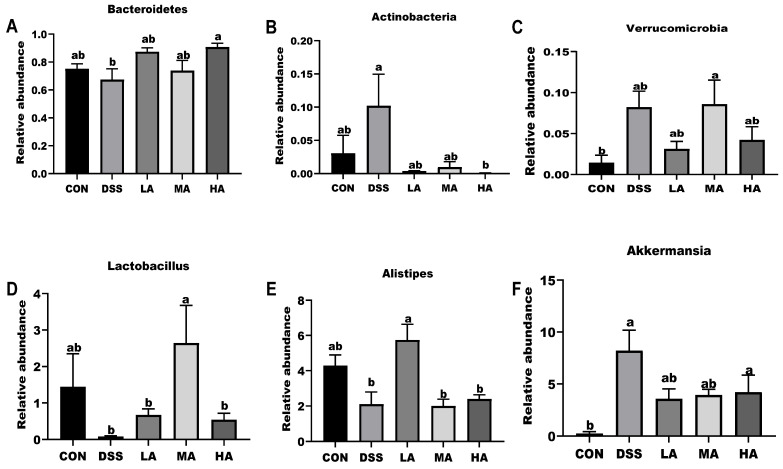
The 16S rRNA bacterial sequences represented in colons. (**A**) The relative abundance of Bacteroidetes at the phylum. (**B**) The relative abundance of Actinobacteria at the phylum. (**C**) The relative abundance of Verrucomicrobia at the phylum. (**D**) The relative abundance of Lactobacillus at the genus. (**E**) The relative abundance of Alistipes at the genus. (**F**) The relative abundance of Akkermansia at the genus. Values are expressed as mean ± SEM, *n* = 6. a, b Mean values with different small letter superscripts mean a significant difference (*p* < 0.05).

**Table 1 nutrients-14-03707-t001:** Standard of the disease activity index (DAI) score.

Score	Stool Consistency	Bleeding	Weight Loss (%)
0	Normal	Normal	0
1	Softer stool/sticks to cage wall	Weak hemoccult-positive spots in stool	1–5
2	Moderate diarrhea/unformed stool	Visual blood in stool/strongly hemoccult-positive stool	6–10
3	Diarrhea (watery stool)	Fresh rectal bleeding	10–15
4	-	-	>15

**Table 2 nutrients-14-03707-t002:** Standard of the histopathological score.

Score	Severity of Inflammation	Crypt Damage	Ulceration
0	Rare inflammatory cells in the lamina propria	Intact crypt and goblet cell	0 foci of ulceration
1	Increased numbers of granulocytes in the lamina propria	Loss of basal 1/3 of crypt and depletion of goblet cells	1–2 foci of ulceration
2	Confluent inflammatory cells extended to submucosa	Loss of basal 2/3 of crypt and depletion of goblet cells	3–4 foci of ulceration
3	Transmural extension of the inflammatory infiltration	Loss of entire crypt and depletion of goblet cells	Confluent or extensive ulceration
4	-	Change in epithelial surface caused by erosion	-
5	-	Confluent erosion	-

**Table 3 nutrients-14-03707-t003:** The primers used for quantitative real-time PCR.

Genes	Primer Sequences (5′-3′)	Serial Number	Product Length/bp
GAPDH	F: GCACAGTCAAGGCCGAGAAT R: GCCTTCTCCATGGTGGTGAA	XM_017321385.2	151
GPX1	F: GGTTCGAGCCCAATTTTACA R: CCCACCAGGAACTTCTCAAA	XM_021172037.2	199
GPX4	F: CTCCATGCACGAATTCTCAG R: ACGTCAGTTTTGCCTCATTG	NM_001367995.1	117
TLR4	F: TTCAGAACTTCAGTGGCTGGATT R: CCATGCCTTGTCTTCAATTGTTT	NM_021297.3	64
MYD88	F: GCATGGTGGTGGTTGTTTCTG R: GAATCAGTCGCTTCTGTTGG	NM_010851.3	108
RIPK3	F: GCCTTCCTCTCAGTCCACAC R: ACGCACCAGTAGGCCATAAC	NM_019955.2	127
RIPK1	F: GCTGTCATCTAGCGGGAGGT R: TCCGCTGTCTAGGTCTGTCT	NM_009068.3	197
MLKL	F: GATTGCCCTGAGTTGTTGCG R: CTCTCCAAGATTCCGTCCACA	XM_030243820.2	89
IL-6	F: AGTTGCCTTCTTGGGACTGA R: TCCACGATTTCCCAGAGAAC	NM_001314054.1	159
TNF-α	F: CTGGGACAGTGACCTGGACT R: GCACCTCAGGGAAGAGTCTG	NM_001278601.1	204

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
