# Peer review of "Improvement of Ulcerative Colitis by Aspartate via RIPK Pathway Modulation and Gut Microbiota Composition in Mice"

_nutrients, 2022, doi:10.3390/nu14183707_

Round 1

Reviewer 1 Report

The present study evaluated Aspartate rule in colitis using DSS mice model. The author mainly focused on genes related to oxidative stress as well as RIP1 and microbiome modulation in various groups. In this manuscript the author claim that ASP alleviate colitis while having influence on the aforementioned factors.  Following parts need to be revised.

1.       In the material and method first section, first its mentioned that various doses were administered orally (Line81-83). Later in line 85-86 PBS and Asp rectal perfusion is mentioned. So now there are two routes of administration.

2.      Line 79 says that water which probably used as a control instead of DSS while in 85 in the rectal perfusion part PBS is mentioned. Either the author has used different things in both the cases and there may have writing issue in both the situation. Also, its not mentioned that either the water/PBS used were tap water or autoclaved.

3.      Tabl3 1. Correct spelling. Stool not stoo.

4.      I guess goblet cell count does include in histopathological score; Please check.

5.      PCOA Plot the taxon of various groups cluster together which indicate that they are closely linked and related. How could it be justified that there is a clear difference in microbiome in terms of beta diversity.?

6.      Presence of More Akkermansia in DSS is a good sign? Also you have mentioned that reduction in the (Line- 390) number of  Akkermansia  was closely related to the occurrence of IBD. Justify your results in figure 7H. Last part of discussion mentioned beneficial microbes, but Akkremansia is not mentioned. Justify.

1.       LDA score figure is not visible. 

Author Response

1 In the material and method first section, first its mentioned that various doses were administered orally (Line81-83). Later in line 85-86 PBS and Asp rectal perfusion is mentioned. So now there are two routes of administration.

Thank you for your review,during first 7 days, we used water as a control while DSS as a colitis model induction. After the model establishing,  we used PBS as a control for Asp was dissolved in PBS. So yes, there are two routes of administration, one is the establishment of colitis model, the other one is Asp treatment period, but unlike the first period, the second period we used rectal perfusion instead of orally administration.

2 Line 79 says that water which probably used as a control instead of DSS while in 85 in the rectal perfusion part PBS is mentioned. Either the author has used different things in both the cases and there may have writing issue in both the situation. Also, its not mentioned that either the water/PBS used were tap water or autoclaved.

Thank you for your review,as we discussed in problem 1, PBS was used for dissolvation of ASP. And thanks for the advice, now we have reedited this part in case of the ambiguity.

3 Tabl3 1. Correct spelling. Stool not stoo

Thank you for your review,stoo has been modified to stool

4  I guess goblet cell count does include in histopathological score; Please check.

Thank you for your review,goblet cell count has been added to the histopathological score.

5 PCOA Plot the taxon of various groups cluster together which indicate that they are closely linked and related. How could it be justified that there is a clear difference in microbiome in terms of beta diversity.?

Thank you for your review,the PCOA Plot has been modified,and the PCOA plot justified that there is a clear difference in microbiome in terms of beta diversity.

6 Presence of More Akkermansia in DSS is a good sign? Also you have mentioned that reduction in the (Line- 390) number of  Akkermansia  was closely related to the occurrence of IBD. Justify your results in figure 7H. Last part of discussion mentioned beneficial microbes, but Akkremansia is not mentioned. Justify.

Thank you for your review,Akkermansia can enhance intestinal barrier function by enhancing the adhesion and integrity of the intestinal epithelium, The abundance of Akkermansia bacteria in each group after DSS treatment was higher than that in the control group, so we guessed that DSS may cause the epithelial cell damage thus damages the gut barrier function, for which increased the abundance of Akkermansia, to fight against the DSS damage.

Reviewer 2 Report

Congratulations to the authors,

I have minor comments:

1) The authors should add how did they assess the data distribution (is the data normal or skewed), what statistical test did they use for that task? Furthermore, they should add that they used a post-hoc test after ANOVA.

2) Sentence with reference num 11 could benefit from an additional reference (or a change, since the paper refered is about mouse male reproductive system)

3) Further English proofreading

4) The pie charts should be more readable or you should discard them (since in this form they are rather messy)

Author Response

1 The authors should add how did they assess the data distribution (is the data normal or skewed), what statistical test did they use for that task? Furthermore, they should add that they used a post-hoc test after ANOVA.

Thank you for your review. The corresponding section has been modified,SPSS statistical analysis histogram was used to analyze the data distribution, and the test data conformed to the normal distribution. At the same time, and add we used a post-hoc test after ANOVA.

2 Sentence with reference num 11 could benefit from an additional reference (or a change, since the paper refered is about mouse male reproductive system)

Thank you for your review. Num 11 has been modified to another relevant  reference.

3 Further English proofreading

Thank you for your review. English has been further proofread.

4 The pie charts should be more readable or you should discard them (since in this form they are rather messy)

Thank you for your review. Pie chart has been deleted.